Machine learning approaches to debris flow susceptibility analyses in the Yunnan section of the Nujiang River Basin

Zhou Jingyi 1
Huang Jiangcheng 2
Sun Zhengbao 3
Yi Qi 1 yiqi1018@126.com
http://orcid.org/0000-0002-6643-7702 He Aoyang 2
1 School of Earth Sciences, Yunnan University , Kunming , China
2 Institute of International Rivers and Eco-Security, Yunnan University , Kunming , China
3 School of Engineering, Yunnan University , Kunming , China
Rossi Mauro
Electronic publication date: 2024 May 20
Publication date: 2024
Volume: 12
Electronic Location ID: e17352
Received 2023 Oct 6; Accepted 2024 Apr 17
Copyright: © 2024 Zhou et al.
Copyright year: 2024
Copyright holder: Zhou et al.
License: This is an open access article distributed under the terms of the Creative Commons Attribution License, which permits unrestricted use, distribution, reproduction and adaptation in any medium and for any purpose provided that it is properly attributed. For attribution, the original author(s), title, publication source (PeerJ) and either DOI or URL of the article must be cited.
License URL: https://creativecommons.org/licenses/by/4.0/

Keywords: Debris flow susceptibility, Random forest model, Support vector machine, Nujiang river, Alpine-valley-area

Funding: National Natural Science Foundation of China (NSFC) 42271086, and 41906148 and 41761109 Yunnan 202301AS070039 Yunnan University KC– 22222780 This work was supported by the National Natural Science Foundation of China (NSFC) under grants No. 42271086, and 41906148, and No. 41761109, the Special project of Basic Research-Key project, Yunnan (No. 202301AS070039) and the Postgraduate Research and Innovation Foundation of Yunnan University (KC– 22222780). The funders had no role in study design, data collection and analysis, decision to publish, or preparation of the manuscript.

==============================
Background

The Yunnan section of the Nujiang River (YNR) Basin in the alpine-valley area is one of the most critical areas of debris flow in China.

Methods

We analyzed the applicability of three machine learning algorithms to model of susceptibility to debris flow—Random Forest (RF), the linear kernel support vector machine (Linear SVM), and the radial basis function support vector machine (RBFSVM)—and compared 20 factors to determine the dominant controlling in debris flow occurrence in the region.

Results

We found that (1) RF outperformed RBFSVM and Linear SVM in terms of accuracy, (2) topographic conditions were prerequisites, and geology, precipitation, vegetation, and anthropogenic influence were critical to forming debris flows. Also, the relative elevation difference was the most prominent evaluation factor of debris flow susceptibility, and (3) susceptibility maps based on RF’s debris flow susceptibility (DFS) showed that zones with very high susceptibility were distributed along the mainstream of the Nujiang River. These findings provide methodological guidance and reference for improvement of DFS assessment. It enriches the content of DFS studies in the alpine-valley areas.

Introduction

Debris flow is a rapid to extremely rapid surge of saturated debris in a steep channel, characterized by strong entrainment of material and water along its path. It is initiated by events like slides, debris avalanches, or rock falls, which can cause sudden undrained loading, leading to liquefaction or increased pore-pressure in the channel bed. As the flow progresses downstream, erosion of steep banks adds to the surge, incorporating more soil and organic debris (Hungr, Leroueil & Picarelli, 2014).

The Yunnan section of the Nujiang River (YNR) Basin, located in the transitional zone between the Kunlun-Qinling Mountains, is the center of southwestern longitudinal ridge and valley area. The terrain is undulating with a relative elevation difference of over 4,700 m (Tang, 2005; Xu, 2016). This region frequently experiences prolonged and intense precipitation during the rainy seasons, increasing the moisture content in the rocky and unconsolidated sediment (Ming, 2006a). Debris flows form under the surface hydrodynamic action and imperil the lives and property of the local population.

On average, there are eight debris flows per 10 km2 in the YNR Basin, which is one of the world’s most severe debris flow areas in China (Tang, 2005; Yang et al., 2017). A total of 283 debris flows occurred in the basin from 1999 to 2008, based on the geological hazard investigation and zoning records of Yunnan Province. In two specific events alone, massive debris flows occurred in Gongshan County on July 26 and August 18, 2010 resulting in nearly 100 deaths and hundreds of millions of Yuan in economic losses (Min et al., 2013).

A clear understanding of the spatiotemporal relationships between debris flows and their evolution factors holds profound implications for society. Firstly, authorities and residents will be able to implement targeted preventive measures by effectively identifying and assessing potential debris flow risk areas, significantly enhancing society’s overall preparedness and ultimately reducing casualties and property losses. Secondly, accurate susceptibility analysis will help avoid construction in potentially hazardous debris flow areas, while spatiotemporal correlation analysis will aid planners in assessing potential impact areas and frequency of debris flows. This will contribute to reducing the impact of disasters on urban infrastructure and enhancing overall resilience of cities. Further, understanding of the spatiotemporal relationships between debris flows and their evolution factors will facilitate more efficient allocation of resources. The proactive deployment of emergency rescue resources ensures a swift and organized response in an event of a disaster, minimizing the overall impact. Additionally, susceptibility analysis serves as the foundation for establishing an effective early warning system. Through monitoring potential debris flow risk areas, timely identification of signs of potential debris flows, and rapid issuance of warnings, residents can take appropriate preventive and evacuation measures, thus maximizing the reduction of casualties. Finally, an in-depth spatiotemporal correlation analysis aids scientists in gaining a better understanding of the formation mechanisms and evolutionary patterns of debris flows, providing a more accurate foundation for risk management (Janizadeh et al., 2019).

Debris flow susceptibility (DFS) assessment methods can be broadly categorized into heuristic approaches, physical models, other conventional methods and data-driven approaches, which have different characteristics and strengths (Table 1; Chang et al., 2019; Dou et al., 2019; Reichenbach et al., 2018; Sun et al., 2021; Zhou et al., 2020). Despite numerous studies addressing the accuracy of various models and the relative merits of predictive outcomes for different areas, comparative research on susceptibility and evaluation factors of debris flows in alpine-valley areas is limited. The scarcity can be attributed to the challenging nature of collecting data in these remote regions, which are characterized by limited transportation, poor road conditions, and inherent difficulties of accessing high mountainous terrain. Additionally, existing studies often rely on conventional debris flow susceptibility assessment methods, which exhibit lower accuracy and fail to meet practical requirements (Liang et al., 2020; Zhang et al., 2019).

Table 1 Classification of DFS assessment methods.

Classification of DFS assessment methods	Instance	Specificities	
Heuristic approaches	Topographical analysis, Precipitation analysis, etc.	Time-consuming.
Unsuitable for large-scale application.
The study results lack comparability due to non-uniform metrics (Dou et al., 2019; Huang et al., 2022).	
Physical models	A shallow water model based on the finite volume method to predict the potential magnitude of debris flows (Bao et al., 2019), Erosion–Deposition Debris flow Analysis (Shen et al., 2018), etc.	They can accurately and efficiently simulate the mechanism of debris flow movement and make predictions. The processes of model building are complex and expensive; assess the susceptibility of single-gully debris flows, rather than larger regions (Luo & Liu, 2018).	
Other conventional methods	Fuzzy logic (Li et al., 2017),
Hierarchical analysis (Liou, Nguyen & Li, 2017),
Network analysis (Sujatha & Sridhar, 2017), etc.	Have limitations in revealing the spatial distribution pattern of non-linearity.	
Data-driven approaches	Support vector machine (Chang et al., 2019),
Random forest model (Liang et al., 2020),
Convolutional neural network (Zhang et al., 2019), Multilayer perceptron (Janizadeh et al., 2019), Boosted regression trees (Xiong et al., 2020), Artificial neural network (Islam et al., 2021), Recurrent neural network (Ngo et al., 2021), Logistic model tree (Chen & Zhang, 2021), Adaptive Boosting (He et al., 2021), Classification and regression trees (Wang, 2021), Generative adversarial networks (Al-Najjar et al., 2021), Long Short-Term Memory (Kavzoglu, Teke & Yilmaz, 2021), etc.	Higher accuracy and more precise prediction results (Oh & Lee, 2017).	

To enhance the accuracy and precision of DFS assessment in alpine-valley areas and explore the key factors influencing debris flow formation along with the spatial distribution map of debris flow classifications, we constructed DFS models based on Random Forest (RF), radial basis function support vector machine (RBFSVM), and linear kernel support vector machine (Linear SVM). This study provides methodological guidance and reference for the improvement of DFS assessment. It also contributes to the enhancement of disaster mitigation and prevention planning in urban and rural areas of the YNR Basin.

Study area

The YNR Basin is located in the longitudinal ridges and valleys belt of western Yunnan Province between 23°07′–28°23′N and 98°07′–100°30′E covering approximately 30,000 km2 (Fig. 1); the area is at the southeastern edge of the strong extrusion zone between the Asian and European plates and the Indian Ocean plate, with strong geological and tectonic movements (Ma, 1999). The geomorphological development is controlled by the deep and large fractures of the Nujiang River, and debris flows are distributed in bands along the fracture zones and gullies (Guo, Luo & Tang, 2015). Study area is located in the southwest monsoon area with distinct wet and dry periods, and the rainy season is concentrated from April to September (Ming, 2006b).

Figure 1 YNR basin (study area).

The YNR basin was divided into the upstream and downstream section by the boundary between Lushui County and Longyang District considering differences in topography and vegetation cover in the study area (Xu, 2016). The upstream area has a typical alpine-valley landscape characterized by high mountains, deep valleys, steep slopes, and swiftly flowing water (Huang et al., 2020a). There are the Gaoligong mountain, the Gawa Gap, the Bilo and Meri snow mountains. The relative height difference reaches 3,000 m between the highest and lowest points of the study area. The Nujiang River is extremely long and narrow in these large mountains, with a maximum basin width of 267 km and a narrowest of only 21 km. The downstream has relatively flat terrain on both sides, with many hills and alluvial fans of uneven sizes.

Vegetation cover of the Yunnan section of the Nujiang River Basin is relatively high, with dominant dry-hot river valley shrub-grassland flanking both sides of the valley. Vegetation types change from an evergreen forest, semi-evergreen forest, deciduous forest, mixed broad-leaved coniferous forest, coniferous forest, and alpine shrubs from valleys to ridges (Luo, 2009; Xu, 2016). The majority of soil types are red-yellow soils with loose texture, poor erosion resistance, and water retention in the basin, and shift in the order of red loam, yellow-red loam, yellow-brown loam, brown loam, dark brown loam, grey-brown forest soil, and alpine meadow soil along with the rising elevation (Liu, 2017).

Rapid socio-economic development of the area accelerates the changes in environment due to human activities such as town and rural built-land expansion, steep slope cultivation, road construction, and mining engineering. Literature indicates that the Nujiang River basin is the most serious geological hazard among the six major basins in Yunnan Province, especially upstream of the YNR Basin (Huang et al., 2020b; Tang & Zhu, 2003).

Methods and data

Research methodology

The overall method flow is shown in Fig. 2.

Figure 2 Research methodology.

(A) General procedure. (B) Detailed procedure of the methods construction.

The Random Forest model

The Random Forest model is effective in capturing and simulating complex nonlinear relationships between debris flows and evaluation factors, and can handle large-scale, high-dimensional debris flow datasets without overfitting. Furthermore, RF indicates the relative importance of each evaluation factor, guiding the understanding of which factors have the greatest impact on debris flow susceptibility. It also demonstrates good adaptability to noise and outliers in the data, making it less susceptible to interference. Therefore, RF exhibits excellent applicability to assessments of debris flow susceptibility (Duan et al., 2022; Zhang & Wu, 2019).

RF is an integrated algorithm consisting of multiple unrelated decision trees that determine the DFS, in which the final output is determined by all decision trees together, and is defined as Breiman (2001),

(1) H(x)=arg⁡maxz∑i=1k⁡I(hi(x)=Z)

where H(x) denotes the results of the model’s predicted DFS for each watershed unit, hi(x) denotes the ith decision tree, x denotes attributes, hi(x)=Z is the prediction of variable Z using the ith tree in variable x, and I(.) is the prediction of each decision tree.

Given a database, RF can be interpreted via the following three steps. Firstly, sample subsets are extracted using the Bootstrap resampling method. Specifically, n sample subsets of the same size as the original sample are extracted using the put-back method. Secondly, a decision tree is constructed for each sample subset. Among the attributes of the sample subset, k attributes are randomly selected. Then, the best partitioning attributes of the nodes between decision trees are selected based on the Gini Index, which is calculated as

(2) Gini(p)=∑k=1k⁡pk(1−pk)=1−∑k=1k⁡pk2

where pk indicates the probability that the selected sample belongs to category k. Smaller Gini Indexes mean that the probability of a selected sample in the set being misclassified is smaller. Finally, n decision trees are combined to generate a random forest (Fig. 3).

Figure 3 The process of the RF model.

Hyperparameters of the RF model can be divided into two categories: those that determine the sampling method, such as bootstrap and the number of classifiers that determine the sampling method, and the number of decision trees. Parameters that determine the decision tree, such as maximum depth (max_depth), minimum number of samples for a leaf node (min_samples_leaf), minimum number of samples required to split an internal node (min_samples_split), the maximum number of features randomly selected as candidates for splitting (max_features), and a criterion that determines the maximum depth, minimum number of samples for a leaf node, minimum number of samples required to split an internal node, the maximum number of features randomly selected as candidates for splitting, and a criterion for the optimal split attribute.

The support vector machine

Support vector machine (SVM) demonstrates superior classification performance on unseen data due to its outstanding generalization capability, laying a crucial foundation for the credibility and practicality of the model in real-world applications. In practical applications, areas prone to debris flows encompass complex environmental features, and SVM’s excellent handling of high-dimensional data provides robust support for assessing susceptibility considering multiple evaluation factors. Additionally, by employing various kernel functions, SVM exhibits flexibility in modeling nonlinear relationships in high-dimensional space, thereby enhancing its applicability in the assessment of DFS. These attributes position SVM as a powerful tool when facing real and complex datasets, providing a reliable analytical framework for accurately evaluating DFS.

There are two cases of linearly divisible and linearly indivisible sample data in the feature space (Fig. 4). As an example of binary data, a binary classification space D(Xi,Yi), i=1,…,l.XiϵRn,Yiϵ{1,−1}, where, l represents the number of samples, and n denotes input dimensionality. The hyperplane ωx+b=0 can be found in the original space when the sample data are linearly divisible separating the two classes of samples completely. When the sample data are linearly indivisible, it is necessary to perform nonlinear mapping Φ(x), mapping it from the input space to a certain feature space, the classification hyperplane can be expressed as ωΦ(x)+b=0; meantime, the optimal hyperplane that requires 2/ω is the largest, and the problem is transformed into a quadratic programming problem, with the application of the Lagrange multiplier method for the solution, namely,

Figure 4 Support vector machine models.

(A) Linearly divisible case. (B) Linearly indivisible case.

(3) {minω22+C∑i=1l⁡εi,s.t.yi(ω⋅xi+b)≥1−εi,εi≥0,i=1,2,...,l

where εi is the slack variable and C is the penalty factor. According to the Kuhn-Tucker (K-T) condition, the following dyadic problem can be obtained:

(4) max∑i=1l⁡ai−12∑i=1l⁡∑j=1l⁡aiajyiyjφ(xi)⋅φ(xj),s.t.0≤ai≤C,∑i=1l⁡aiyi=0

By solving the dyadic problem of this quadratic programming, the discriminant function is obtained as:

(5) f(x)=sign∑i=1l⁡aiyi[Φ(xi)⋅Φ(xj)]+b

According to the relevant theory of generalized functions, as long as a kind of kernel function K(xi,yi) satisfies the Mercer condition, it corresponds to an inner product in a certain transformed space, K(xi,yi) = ϕ(xi)⋅ϕ(xj), and linear classification of a certain nonlinear transformation can be achieved by using a different inner product function in the optimal classification surface (Suykens & Vandewalle, 1999).

Linear kernel, polynomial kernel, sigmoid kernel, and the radial basis function (RBF) are commonly used in support vector machine models, where

Linear kernel

(6) K(y,y′)=yTy′

RBF

(7) K(y,y′)=exp⁡(−12σ2y−y′2)

where y and y′are both basis vectors in the feature space, and σ is a model’s hyperparameter. Compared with the linear kernel, the RBF can transform the features dimensionality for reducing computational complexity, which is extremely suitable for predicting DFS in high-dimensional feature spaces (Lin & Lin, 2003). The penalty parameter C, an empirical parameter in the SVM model, is used to control the tolerance of systematic outliers, allowing for a few outliers to exist in the opponent classification. A higher value of the penalty parameter leads to fewer outliers in the opponent classification. What’s more, the radial basis function kernel has an additional kernel parameter γ i.e., kernel bandwidth to be optimized, where γ = 1/2σ2. As γ increases, the fit changes towards non-linear.

Accuracy evaluation metrics

Evaluation of model’s accuracy is critical for decision-makers and relevant institutions. High-precision model predictions contribute to more precise decision-making, ensuring that measures taken are scientifically sound and effective. Comparing the accuracy of different models in practical applications contributes to the selection of the most suitable model for a given region or topography, thereby bolstering the credibility of predictions. Additionally, accuracy evaluation provides feedback on the current model performance, guiding continuous improvement efforts.

Accuracy, Precision, Recall, Kappa, F1-score, receiver operating characteristic (ROC) curve, and area under ROC (AUC) are used as evaluation metrics. Accuracy represents the proportion of correctly classified debris flow samples, serving as a key indicator for overall model performance. Precision refers to the proportion of samples with positive cases that are correctly predicted which are critical for reducing false positives and ensuring the rational use of limited resources. Recall indicates the proportion of positive cases that are correctly predicted in the true sample, and is essential for minimizing the risk of overlooking potential hazard zones. F1-score offers a balanced assessment of precision and recall, guiding the establishment of reasonable warning and management strategies. A higher F1-score indicates greater model accuracy. Kappa measures model consistency, indicating its ability to make similar judgments under different conditions or at different times. In the context of dynamic changes in debris flow risk, model stability is essential for providing continuous and effective risk assessments. A higher Kappa means greater classification accuracy. The ROC curve aids decision-makers in balancing sensitivity and specificity by using true positive rate (TPR) as the vertical axis and false positive rate (FPR) as the horizontal axis. The AUC, obtained by integrating the ROC curve, reflects the model’s classification effectiveness. Even in situations with imbalanced positive and negative samples, a higher AUC value indicates superior model accuracy. Their definitions are as follows,

(8) Accuracy=TP+TNTP+FP+TN+FN

(9) Precission=TPTP+FP

(10) Recall=TPTP+FN

(11) F1−score=2∗Precision∗RecallPrecision+Recall

(12) Kappa=Accuracy−pe1−pe

(13) pe=(TP+FP)∗(TP+FN)+(FN+TN)∗(FP+TN)(TP+FP+FN+TN)∗(TP+FP+FN+TN)

The ROC curve is generated using the true positive rate (TPR) as the vertical axis and FPR as the horizontal axis. The AUC is obtained by integrating the ROC curve and it reflects the classification effect of the model. The closer its value to 1, the better and more accurate the model. The calculation is as follows,

(14) TPR=TPTP+FN

(15) FPR=FPFP+TN

where in Eqs. (8)–(15), TP stands for the true positive rate, FP represents the false positive rate, TN signifies the true negative rate, and FN denotes the false negative rate.

To facilitate horizontal comparisons of model predictions, many studies use the equal spacing method to classify DFS into five zones and confirm the method’s applicability (Liang et al., 2020; Liu, Miao & Tian, 2017). Therefore, in this study, we divided the predicted debris flow susceptibility calculated by the three models into five classes, from low to high, corresponding to the very low susceptibility zones (0 to <0.2), low susceptibility zones (0.2 to <0.4), medium susceptibility zones (0.4 to <0.6), high susceptibility zones (0.6 to <0.8) and very high susceptibility zones (0.8 to <1), respectively (Zhang & Wu, 2019).

Data and processing

Evaluation units

Raster cells and watershed units in DEM are commonly used for susceptibility assessments (Zou et al., 2017). Raster cells are more convenient for modelling and calculation because of their regular shape and uniform size, while watershed units can represent integrated geomorphological characteristics of hydrological processes, and that helps in obtaining the actual conditions of debris flow (Liu et al., 2018; Qiang et al., 2019; Zhang et al., 2022). Therefore, we adopted watershed units as the basic evaluation units and used ArcMap’s hydrological analysis tools to categorize the 30 m spatial resolution DEM data of the YNR Basin. Finally, the study area was divided into 1,070 watershed units (Esri, 1981).

Evaluation factors

The formation of debris flows is determined by a combination of factors (Huang et al., 2022). After conducting a thorough investigation, data collection, and preliminary analysis of existing data on the historical background, geological structure, topography and geomorphology, hydro-meteorology, soil and vegetation, and human activities in the study area, we selected evaluation factors from five aspects: topographic, rainfall, geological, and vegetation conditions, and human activities (Table 2).

Table 2 Selected evaluation factors and their data sources.

Category	Evaluation factors	Data source	Resolution	
Topographic conditions	Relative elevation difference	2009 GDEMV3 digital elevation data from the website (https://www.gscloud.cn/).		
Average slope	30 m	
Watershed area		
Rainfall conditions	Average rainfall during the rainy season	Data from NASA 2020 GPM day-by-day precipitation.		
Number of heavy rainstorms	1° × 1°	
Number of rainstorms		
Number of heavy rains		
Geological conditions	Fracture zone density	Data of 1995 from the website (https://www.resdc.cn/).	Vector data	
Sand content	30 m	
Silt content	
Clay content	
Vegetation conditions	NDVI	2020 vegetation cover data are derived from the website (http://www.nesdc.org.cn/).	30 m	
Human activities	Land use	2020 land use data is derived from the website (https://mulu.tianditu.gov.cn/commres.do?method=globeIndex).	30 m	
Railway density	Data from the 2020 Gaode map.	Vector data	
Highway density	
Density of urban quaternary roads	
Density of urban tertiary roads	
Density of urban secondary roads	
Density of urban primary roads	
Density of county and town roads	

(1) Topographic conditions. Topographic conditions are critical to the formation of debris flows. We chose relative elevation difference and slope to represent potential energy of watersheds and the ability to carry rocky soil, respectively, and watershed area for runoff and sediment yield calculations based on previous research in this area by Liu & Tang (1995), Sun et al. (2021). Therefore, based on the DEM with a spatial resolution of 30 m, the mean relative elevation difference and average slope of each watershed unit were calculated using the ArcMap function of zonal statistics as a table, and the area of each watershed unit was extracted using the “calculate geometry” function (Esri, 1981).

(2) Rainfall conditions. Rainfall is necessary for debris flow incubating and triggering (Cui, Yang & Chen, 2003; Liu, Miao & Tian, 2017; Xu, 2016). Antecedent rainfall serves mainly to wet or soften the soil and reduce the stability of rocky soil. Short-duration heavy rainfalls create a strong mechanical impact on the soil that is almost saturated, and disrupt the equilibrium of the slope causing debris flows (Pan et al., 2012; Tan, Yang & Shi, 1990; Zhang & Guo, 2021). To address the effect of rainfall, we selected three factors to characterize the triggering effect of heavy rainfall on debris flows in 2020: the number of heavy rainstorms, the number of rainstorms, and the number of heavy rains. Based on the distinctive interannual variability in precipitation distribution in the study area between dry and rainy seasons, we chose the average rainfall during the rainy season (April to September) to characterize the effect of antecedent precipitation on developing debris flows. Therefore, a total of 183 daily precipitation data were extracted for dates from 2020/04/01 to 2020/09/30 in the study area, and the field calculator and the regional statistical function of ArcMap was used to calculate the average rainfall for the 2020 rainy season for each watershed unit. Using ArcMap’s model builder (Esri, 1981), a total of 366 daily precipitation data for 2020 were sequentially screened for the number of heavy rainstorms with a cumulative daily precipitation of 100–250 mm, the number of rainstorms with 50–100 mm, and the number of heavy rains of 25–50 mm; then, we summed the number of days in compliance with the raster cells using the field calculator, and calculated the average number of heavy rainstorms, rainstorms, and the heavy rainfall for each watershed unit with the ArcMap function of zonal statistics table (Esri, 1981).

(3) Geological conditions. Fracture zones affect the continuity and stability of rocky soil, and surface soil provides a rich sediment source for debris flows (Pham et al., 2016). Consequently, we used fracture zone density and soil texture to characterize the influence of geological conditions on debris flows. Fracture zone density was calculated by dividing the length of the fracture zone in each watershed unit by the watershed area. Soil texture was calculated separately using the average content of clay, silt, and sand within the watershed unit and the function of zonal statistics.

(4) Vegetation conditions. Plant roots stabilize the rock and soil mass, and increase soil resistance to erosion (Huang et al., 2022). To some extent, plant roots inhibit erosion and hinder the sliding of the topsoil (Zhao, Wu & Wang, 2006). We used the ArcMap function of zonal statistics to calculate the average Normalized Difference Vegetation Index of each watershed unit to characterize vegetation cover (Esri, 1981).

(5) Human activities. Geological structure and surface become unstable with land use changes, which also generate loose deposits providing material sources for debris flows (Huang et al., 2022; Tien Bui et al., 2017; Xu, 2016). Road construction can indicate regional land development intensity. We used the ArcMap function to calculate land use types with the largest proportions for each watershed unit. Densities of highway density, railway, urban primary roads, urban secondary roads, urban tertiary roads, urban quaternary roads, and county and town roads were calculated using corresponding road type lengths divided by the watershed area (Esri, 1981).

Data pre-processing

We used the Random Forest (RF) model to reduce the noise in the data sets and to select evaluation factors following the approach of Kursa & Rudnicki (2011).

First, a RF model was built with all the evaluation factors in Python after digitizing, data formatting, and unifying georeferencing. Second, the model was trained again using the GridSearchCV module (Cournapeau, 2007b), which iterates through all permutations of incoming parameters to find the best hyperparameter. Third, contributions of the evaluation factors were obtained and ranked. Finally, evaluation factors were filtered. Through analyzing the initial factors and their contribution in Table 3, we determined that the contribution of six factors, namely railway, highway density, the density of urban primary roads, the density of urban secondary roads, the density of urban tertiary roads, the density of urban quaternary roads, were less than 0.01%, and they apparently did not in the same order of magnitude as other factors. Therefore, we considered these six factors as noisy data and removed them. From further evaluation. The remaining evaluation factors are shown in Fig. 5.

Table 3 Contribution of evaluation factors before and after pre-processing.

No.	Initial factors and their contribution	Processed factors and their contribution	
Evaluation factors	Contribution	Evaluation factors	Contribution	
1	Relative elevation difference	0.184	Relative elevation difference	0.274	
2	Average slope	0.121	Watershed area	0.087	
3	Average rainfall during the rainy season	0.092	NDVI	0.074	
4	Number of heavy rainstorms	0.073	Density of county and town roads	0.072	
5	Clay content	0.069	Average slope	0.064	
6	Number of rainstorms	0.068	Average rainfall during the rainy season	0.061	
7	NDVI	0.057	Land use	0.061	
8	Silt content	0.057	Sand content	0.057	
9	Number of heavy rains	0.055	Silt content	0.050	
10	Sand content	0.053	Clay content	0.049	
11	Watershed area	0.052	Number of heavy rainstorms	0.044	
12	Fracture zone density	0.051	Fracture zone density	0.039	
13	Density of county and town roads	0.048	Number of heavy rains	0.036	
14	Land use	0.020	Number of rainstorms	0.032	
15	Railway density	0			
16	Highway density	0			
17	Density of urban quaternary roads	0			
18	Density of urban tertiary roads	0			
19	Density of urban secondary roads	0			
20	Density of urban primary roads	0			

Figure 5 Processed factors affecting debris flows susceptibility.

(A) Relative elevation difference. (B) Average slope. (C) Watershed area. (D) Average rainfall during the rainy season. (E) Number of heavy rainstorms. (F) Number of rainstorms. (G) Number of heavy rains. (H) Fracture zone density. (I) Sand content. (J) Silt content. (K) Clay content. (L) NDVI. (M) Land use. (N) Density of county and town roads.

(1) Data annotation. The debris flow inventory was derived from the nationwide geohazard census done by the Resource and Environment Science and Data Center (www.resdc.cn/). After data verification with local histories, books, reports, statistic data, relevant field surveys, and related literature, a total of 274 debris flow hazards in the study area were found as of the end of 2019, and the attribute information included the field number, geographic location, damage, groundwater grade, and the current degree of stability, and so on. We used this inventory to annotate each watershed unit. Those units that had experienced debris flows were assigned a label of ‘1’, while those that had not were assigned a label of ‘0’. The entire set of watershed units was subsequently divided into two groups, namely ‘debris flow’ and ‘no debris flow’.

(2) Data sampling. To prevent sample imbalance from affecting model accuracy, we used the synthetic minority oversampling technique (SMOTE) to balance the sample size, which analyzed a small number of data and added simulated new data to the dataset when needed (Wu, Yang & Niu, 2020; Zhu & Zhang, 2022). Eventually, the watershed unit ratio with ‘debris flow’ to those with ‘no debris flow’ was 1:1, and the total sample size was 2,140.

(3) Datasets division. The whole dataset was divided into two subsets of 70% to 30% for DFS model training and testing, respectively (Huang et al., 2022).

Experiments and analysis of results

Model construction

The RF and SVM models were trained by using the scikit-learn library in Python, which integrates various machine-learning methods. First, RF was generated using the Random Forest Classifier method (Cournapeau, 2007a), and Linear SVM and RBFSVM were generated using the SVC method (Cournapeau, 2007c). Second, we adjusted the values of the hyperparameters, taking into account the relationship between model complexity and generalization error, to minimize the generalization error and improve the accuracy and generalization ability of the model (Duan et al., 2022). The order of adjusting hyperparameter values for RF model based on their effects on model complexity was as follows: the number of classifiers, maximum depth, the minimum number of samples of the leaf nodes, the condition limiting continuation of the subtree division, the maximum number of features, and the decision tree algorithm. Specifically, we employed the ten-fold cross-validation method to generate learning curves for each hyperparameter within a large interval (Cournapeau, 2007c). Once we determined the range of subintervals with the highest accuracy, we used the grid search method to determine the optimal values. As the SVM model had fewer hyperparameters, we adjusted the values of it only using the grid search method (Cournapeau, 2007b). The search range and hyperparameter values for each model are shown in Table 4.

Table 4 Hyperparameter values for each model.

Model	Search range	Hyperparameter value	
RF	(0, 200, 1)	the number of classifiers = 158	
(1, 40, 1)	the maximum depth = 21	
(1, 51, 1)	the minimum number of samples of the leaf nodes = 1	
(2, 51, 1)	the condition limiting the continuation of the subtree division = 2	
(3, 15, 1)	the maximum number of features = 11	
Linear SVM	[0.1, 1, 10, 100, 1,000]	C = 1	
RBFSVM	[0.1, 1, 10, 100, 1,000]	C = 5	
(0.001, 0.1, 0.001)	Gamma = 0.05	
Note:

The first, second, and third digits in “()” represent the minimum value, maximum value, and step size, respectively. The numbers in “[]” represent all attempted values.

DFS models for the YNR Basin were developed based on the optimal hyperparameters for RF, Linear SVM, and RBFSVM methods. In response to the characteristics of alpine-valley areas with distinct wet and dry conditions and complex debris flow genesis in the context of geographic big data, and based on the applicability of the RF and the SVM models in high-dimensional, non-linear data, the model used average rainfall during the rainy season to assess the impact of rainfall on debris flow and used RF and SVM to quantitatively assess the drivers and study area susceptibility.

Analysis of results

Analysis of the pre-processing results

The AUC of the RF model using test data increased from 0.73 to 0.97. The ROC curve was closer to the upper left corner (Fig. 6). Further, model training time decreased by 23%, from 105 to 84 s.

Figure 6 ROC curves.

Data preprocessing eliminated the impact of redundant data on the model and on the remaining evaluation factors, so the contribution rate of the remaining evaluation factors such as topographic conditions, human activities, and vegetation conditions increased by 0.068, 0.065, and 0.017, respectively, while those of rainfall and geological conditions decreased by 0.115 and 0.035, respectively (Fig. 7). In addition, the contribution rate of the most important evaluation indicator (relative elevation difference) increased by 0.09 (Table 3).

Figure 7 The contribution rate of the five main categories of evaluation factors before and after pre-processing.

Analysis of evaluation factors

Experimental results indicated that topographic conditions were the decisive factors during the formation of debris flow in the YNR Basin, and geological conditions, rainfall conditions, human activities, and vegetation conditions were important factors, with contribution rate corresponding to 0.425, 0.195, 0.173, 0.133, and 0.074 (Fig. 7), respectively. The top three explanatory conditions were topographic, geological, and rainfall.

The relative elevation difference (contribution rate: 0.274) was the most important evaluation indicator with a key role in the formation of debris flows. This was followed by watershed area, NDVI, and density of county and town roads ranking 3rd and 4th, respectively. The others were in the order of average slope, average rainfall during the rainy season, land use, sand content, silt content, clay content, the number of heavy rainstorms, fracture zone density, and the number of heavy rains. The number of rainstorms had a lesser impact on the formation of debris flows (Fig. 8).

Figure 8 The contribution of evaluation factors to the total variability in debris flow formation.

Model accuracy analysis

The RF model had higher values of Accuracy, Precision, Recall, F1-score, and Kappa (Table 5), and its ROC curve converged faster than that of RBFSVM and Linear SVM (Fig. 9). This indicated that RF was most suitable of the three models for examination of the spatial correlation between historical debris flows and elevation factors, improving the assessment accuracy of DFS.

Table 5 Comparison of model accuracy.

Model	Accuracy	Precision	Recall	F1-score	Kappa	
RF	0.91	0.91	0.94	0.92	0.83	
RBFSVM	0.89	0.90	0.87	0.89	0.79	
Linear SVM	0.68	0.64	0.77	0.71	0.37	

Figure 9 The ROC curves.

Susceptibility analysis

Susceptibility class and density distribution of debris flows were positively correlated with a higher density of debris flows leading to higher susceptibility classes. The difference in density between very high and very low susceptibility zones serves as an indicator of the predictive performance of the model (Li et al., 2022). Our results showed that the prediction performance of RF was better than that of RBFSVM, while that of linear SVM was the lowest (Fig. 10). All three models demonstrated an increase in debris flow density with increasing susceptibility class but the difference in debris flow density between the very high and very low susceptibility zones was greatest for the RF model with 47 debris flows per 1,000 km2, followed by RBFSVM with 37 debris flows per 1,000 km2, and Linear SVM with 11 debris flows per 1,000 km2. These results indicated that the RF model was more adept at discriminating very high and high susceptibility zones and exhibited superior predictive performance when compared to the other two models.

Figure 10 Density of debris flows in susceptibility zones of the three models.

In addition, we used prediction rate curves to visualize the model’s predictive performance. On the X-axis, we sorted the predicted susceptibility of the test dataset in descending order (i.e., from large to small susceptibility), using the method of accumulation of the corresponding watershed area; The cumulative proportion of the number of debris flow in the corresponding watershed unit was represented on the Y-axis. The farther the verification rate curve was from the diagonal from 0 to 1, the better the predictive power of the model. The larger the gradient in the first half of the curve, the stronger the prediction ability (Chung & Fabbri, 1999; Remondo et al., 2003). As can be seen from Fig. 11, the RF model had the strongest prediction ability, followed by RBFSVM, and Linear SVM was the lowest.

Figure 11 Prediction rate curves.

Based on the analysis of historical debris flows and the overlap in susceptibility zones (Fig. 12), we determined that RF and RBFSVM modelled susceptibility zones in the YNR Basin better than the linear SVM. The linear SVM predicted many of the very high and high susceptibility zones which had no historical debris flows, therefore the credibility of its predictions was low. The high dimensionality of debris flow data may be a contributing factor, as linear SVM had difficulty effectively capturing the complex nonlinear relationships in the dataset. This resulted in tendency to overfit the training data, reducing its accuracy in predicting susceptibility to debris flow hazards. The Kappa value of the linear SVM model was only 0.37 (Table 5), indicating that the model was unable to make consistent predictions at different conditions or times, indirectly confirming the existence of the overfitting problem. This instability limits the applicability of the model to DFS prediction.

Figure 12 Susceptibility zoning in the upstream section of the YNR Basin.

(A) RF model. (B) RBF SVM model. (C) Linear SVM model.

The DFS spatial distribution map obtained with RF showed that the very high susceptibility zones in the upstream section of the YNR Basin were mainly distributed along the mainstream of the Nujiang River. The dominant factors were topographic and geological conditions in the very high susceptibility zones of Gongshan, Fugong, and Lushui counties associated with active neotectonic movements resulting in the development of numerous bulge structures and compressional folds. The swiftly flowing water, driven by the effects of tectonic activity and fluvial erosion, results in the formation of steep riverbanks. After the Holocene, the bulge of the mountains and the deepening of the river valleys led to the formation of alpine-valley landscapes. Furthermore, the deep and large fractures of the Nujiang River, along with numerous tectonic fractures and joint fissures, control the geomorphological development in the area, leading to extreme gravity geomorphology, including debris flows (Liu & Tang, 1995; Tang & Zhu, 2003).

The RF model performed more robustly for susceptibility zoning in the YNR Basin (Fig. 13), as shown by the overlap between historical debris flows and susceptibility areas and by the model accuracy. The modeling results of RBFSVM showed an overall higher susceptibility than that indicated by historical data. In the Linear SVM, many historical debris flow areas were distributed in very low and low susceptibility zones and few historical debris flow areas were distributed in some high susceptibility zones, which rendered predictions less reliable, lead to overfitting of the training data, and resulted in lower accuracy in predicting the susceptibility to debris flow disasters.

Figure 13 Susceptibility zoning in the downstream section of the YNR Basin.

(A) RF model. (B) RBF SVM model. (C) Linear SVM model.

Consequently, the RF model was used for susceptibility zoning in the downstream section of the YNR Basin. The results showed that areas with very high and high susceptibility to debris flows were concentrated in northern Longling County, northern Longyang District, eastern and southeastern Zhenkang County, eastern Shidian County, and central Yongde County, and those of susceptibility probability response to 0.86, 0.85, 0.84, and 0.81, respectively.

Discussion

This study demonstrated applicability of the RF model to assess DFS in the YNR Basin. The RF model exhibited a comparable AUC to that obtained with the Backpropagation Neural Network and had higher Accuracy and a greater overlap between the predicted very high and high susceptibility zones and historical debris flows (Wang et al., 2010). In addition, the difference in AUC, very high and very low susceptibility zone debris flow density for the RF model outperformed the deterministic coefficient model (Li, Yang & Wei, 2019). Further, the methods in the RF model do not rely on expert experience, making it more objective and accurate than the numerical division of the sensitivity of each factor (Tang, 2005) and the method of assigning different weights to each factor (Tang, 2005). We also further validated the inferences of Tang and Li and others regarding more detailed results (Li, Yang & Wei, 2019; Tang, 2005). Nevertheless, the distribution of very high and high susceptibility zones in this study aligned with historical debris flows better and had a higher AUC than that exhibited in the study of Li, Yang & Wei (2019), Tang (2005), indicating a higher reliability of susceptibility zone prediction in this study. Finally, Guo, Luo & Tang (2015) and Xu (2016) showed that topography, source of debris flow, and rainfall were the three main conditions for forming debris flows in Nujiang River Basin, reflecting the ranking of debris-flow contributing factors in this study.

Data pre-processing based on the contribution rate of factors generated by the RF model was reliable. Multicollinearity rarely influences the proposed model due to a large number of training samples. Thus, we used RF to reduce noise, which had a great effect on the accuracy of the model, and the accuracy was significantly improved. This is possible because nodes of the RF model are randomly selected with equal probability to construct decision trees; the evaluation factors with a lower contribution may negatively affect model performance and increase generalization error (Rogers & Gunn, 2006). Furthermore, the proposed model measures the importance of the evaluation factors in terms of their ability to contribute to predicted results in proposed model rather than exclusively in terms of their contribution to the predicted results (Zhang et al., 2019). In addition, higher AUC and a better-performing ROC curve further validate that eliminating noise with RF leads to more accurate results.

Pre-processing removed six evaluation indicators. Variables with a higher number of categories in the RF will tend to contribute more; 80% of the existing roads in the study area are roads below Class IV; the relatively few railways, highways, urban quaternary roads, urban tertiary roads, urban secondary roads, and urban primary roads result in a low number of categories for these six evaluation factors. Additionally, the number of rainstorms in the dataset was relatively sparse, resulting in small information gain during decision tree generation. Thus, its contribution is low.

Model applicability. (1) The RF and RBFSVM models are well suited to DFS assessments which require high-dimensional data; in addition, related literature shows that they have higher accuracy in landslide, flood, and other disaster susceptibility assessments than other models (Fang et al., 2022; Prasad et al., 2022). The linear SVM model cannot capture complex nonlinear relationships well and is prone to overfitting, thus not applicable to DFS assessment. In this study, the linear SVM model predicted many zones with very high and high debris flow susceptibility but without historical debris flow distribution, therefore, its prediction credibility was low. (2) The RF and RBFSVM models have some limitations. Training the RBFSVM model with high-dimensional data is time-consuming, sensitive to noise and outliers, and demands meticulous data cleaning and filtering. Additionally, predictions of the RBFSVM, relying on nonlinear mapping, are relatively challenging to interpret. On the other hand, the RF model tends to favor lower contributions when presented with sparse data. Six evaluation indicators were removed in the preprocessing portion of this study, but they may, in fact, contribute to the formation of debris flow; The results of the quantitative ranking of factor contributions to DFS in the study area need to be investigated further. (3) This study clarifies the applicability, accuracy, and limitations of the three models, providing researchers with methodological references and directions for model improvement; in addition, it provides scientific basis for disaster prevention and avoidance in the alpine-valley area.

Elevation factors of debris flows had dissimilar effects at different scales. At the scale of the Yunnan section of the Nujiang River Basin, the major evaluation factors for debris flows were topographic, geological, and rainfall conditions. However, the main evaluation factors may change when scaled down to each county (Cheng, Yu & Chang, 2010). Thus, precipitation significantly impacts Longling, Zhenkang, Shidian, and Yongde Counties, resulting in an increased likelihood of debris flows, floods, and other disasters. Longling County, Longyang District, and Zhenkang County experience extensive human activities, including construction projects for water conservation, road construction, mining, and logging, leading to severe soil erosion. Geology also significantly affects these areas, such as Longling County, which features a large number of fractures on the eastern side of Chongshan; Zhenkang County has fractures and folds in varying directions contributing to a complex and variable geological structure. In addition, Yongde County has a high concentration of mudstones, sandstones, and slate with looser textures and more developed joint fissures (Cheng et al., 2000; Hou et al., 2005; Wu, Li & Qian, 1993; Xiao & Wu, 1992; Zhang & Li, 1997). Analysis of the evaluation factors of debris flows must be tailored to different regional scales in complex terrain areas.

This study tested the underlying techniques of the model for evaluating the susceptibility of alpine-valley-areas to debris flows, which has an application value for similar research at watershed scales. However, there are some limitations that need to be further improved. First, we did not consider the temporal distribution of debris flows in the model due to insufficient data. Treating debris flows occurring at different times as a type of marker sample would affect the accuracy of the cause analysis of debris flows (Huang et al., 2022). Second, the assessment of the strengths and weaknesses of the predictions was based on the assumption that susceptibility class and the density of debris flow are correlated (Li et al., 2022). Extensive field research in future studies, combined with modeling will significantly improve reliability of the results.

Conclusions

Data acquisition for debris susceptibility assessments in the Yunnan section of the Nujiang River (YNR) Basin is challenging; the predominant use of traditional debris flow susceptibility assessment methods results in low accuracy that fails to fulfill practical needs. Here, we addressed these issues with systematical collection and processing of relevant debris flow data, including an analysis of the performance of three machine learning algorithms in analyzing debris flow susceptibility in the alpine-valley area. The results indicated that the RF model outperforms both the RBFSVM and the linear SVM models in terms of accuracy and precision of prediction indicating that the RF model is more suitable for susceptibility assessment of debris flow in the YNR Basin. This study provides valuable methodological analysis and directions for improvement of the model.

The contribution rate of the evaluation factors generated by the RF model showed that topographic conditions were the decisive factor in the formation of debris flows in the YNR Basin. Geology, rainfall conditions, human activities, and vegetation conditions are essential to forming debris flows. In addition, the relative elevation difference was vital among the 20 evaluation factors in the formation and occurrence of debris flows in our study area.

The results of the RF-based DFS classification distribution map indicated that the very high susceptibility zones are mainly distributed along the mainstream of the Nujiang River. Very high susceptibility zones are primarily situated in Gongshan County, Fugong County, Lushui County, northern Longling County, northern Longyang District, eastern and southeastern Zhenkang County, eastern Shidian County, and central Yongde County where the terrain and geological conditions are extremely conducive to the development of gravity geomorphology. These results support efforts in implementing more targeted preventive measures in very high and high susceptibility zones, significantly enhancing overall preparedness for debris flows.

Additional Information and Declarations

Competing Interests

Author Contributions

Data Availability

The authors declare that they have no competing interests.

Jingyi Zhou conceived and designed the experiments, performed the experiments, analyzed the data, prepared figures and/or tables, authored or reviewed drafts of the article, and approved the final draft.

Jiangcheng Huang conceived and designed the experiments, performed the experiments, analyzed the data, prepared figures and/or tables, authored or reviewed drafts of the article, and approved the final draft.

Zhengbao Sun conceived and designed the experiments, performed the experiments, analyzed the data, prepared figures and/or tables, authored or reviewed drafts of the article, and approved the final draft.

Qi Yi conceived and designed the experiments, performed the experiments, analyzed the data, prepared figures and/or tables, authored or reviewed drafts of the article, and approved the final draft.

Aoyang He conceived and designed the experiments, performed the experiments, analyzed the data, prepared figures and/or tables, authored or reviewed drafts of the article, and approved the final draft.

The following information was supplied regarding data availability:

The code is available at Zenodo: JYzhouzhou. (2024). JYzhouzhou/peerj_DFS: First release of my code (v1.0.1). Zenodo. https://doi.org/10.5281/zenodo.10516748.

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
