# Peer review of "Machine learning approaches to debris flow susceptibility analyses in the Yunnan section of the Nujiang River Basin"

_PeerJ, doi:10.7717/peerj.17352_

## Round 0.1 · original submission · Major Revisions

· Academic Editor

Major Revisions

The manuscript proposes machine learning (ML) models to derive ‎the susceptibility posed by debris flows in the Nujiang River basin. Both the reviewers have decided for a major revision, highlighting important elements to address in the analyses. I kindly suggest accounting for them carefully, giving proper answers and solutions. In line, with the reviewer's comment, I also kindly ask to deepen the data description part on landslide data sample collection and characterization and to highlight/characterize better the manuscript novelty with respect to the literature. In line with the reviewers, I confirm that a major revision is needed before the manuscript's acceptance.

**Language Note:** The review process has identified that the English language must be improved. PeerJ can provide language editing services - please contact us at [email protected] for pricing (be sure to provide your manuscript number and title). Alternatively, you should make your own arrangements to improve the language quality and provide details in your response letter. – PeerJ Staff

Reviewer 1 ·

Basic reporting

Dear Editor,‎

I appreciate the opportunity to revise the manuscript titled "Machine learning ‎approaches for debris flow susceptibility analyses in Yunnan section of the ‎Nujiang river basin," which was submitted to "PeerJ" The manuscript focuses ‎on utilizing several comparative machine learning (ML) classifiers on flow ‎susceptibility analyses (FSA) for Nujiang river basin. While the topic is ‎intriguing, there are several areas in which the manuscript requires ‎improvement. In light of this, I kindly request that the authors consider the ‎following comments:‎
General: ‎
‎-‎ Considering the existing literature, the motivation of the manuscript is ‎not convincing. There is ‎a large literature associated with ML application ‎on FSA, and various techniques ‎have been applied/developed to assess ‎the susceptibility better. I expect to see a novelty in a ‎manuscript dealing ‎with FSA because we already have many case studies. I ‎do not think that ‎new case studies would help the scientific community to go one step ‎further. Could you please stress on your contributions.‎
‎-‎ The paper is generally readable in English. However, I would recommend ‎that the authors have it reviewed by a fluent English speaker to eliminate ‎any potential errors or issues.‎
‎-‎ The paper needs a stronger and clearer introduction that emphasizes the ‎significance and real-world applications of FSA. Why is FSA important? ‎What are the implications of accurate FSA for the study area, in this ‎case? Providing a strong motivation at the beginning can help readers ‎understand the relevance of the study.‎

Technical: ‎
‎-‎ The concluding remarks of the abstract are not well-written. It’s merely ‎the repetition of the ‎objectives and title of the manuscript.‎
‎-‎ Incorporate a literature review of the methods developed and applied in ‎ML applications for FSA within the "Introduction" section.‎
‎-‎ Highlight the current challenges and limitations in obtaining FSA data to ‎underscore the significance of the applied methods.‎
‎-‎ Consider presenting a table that highlights the advantages and ‎disadvantages of these methods for clarity. Towards the conclusion of ‎this section, emphasize the superiority and reiterate the novelty of your ‎work.‎
‎-‎ Please provide the table of hyper-parameters values of all algorithms‎.‎
‎-‎ Incorporate a subsection within the "Discussion" section to explicitly ‎outline the primary limitations, broader applicability of your methods, ‎and the implications of your findings.‎
‎-‎ The methodology section should provide more details on each methods. ‎Explain how each method works, how they were applied in this study, ‎and the reasons for choosing these specific methods. This will enhance ‎the paper's accessibility to readers from various backgrounds.‎
‎-‎ The paper provides percentages for splicing accuracy and global feature ‎segmentation overlap rate. However, it would be helpful to explain the ‎significance of these numbers. What do they mean in the context of FSA? ‎How do these results compare to other existing methods?.‎
‎-‎ Summarize the key findings and contributions of your research in the ‎conclusion section. What are the main takeaways from your study, and ‎how do they advance the field of FSA?‎
‎-‎ Including figures or diagrams to illustrate the methodology or results can ‎greatly enhance the paper's accessibility and help readers grasp complex ‎concepts.‎


Best regards,

Experimental design

.

Validity of the findings

.

·

Basic reporting

This manuscript presents debris flow susceptibility prediction in the Yunnan section of the Nujiang River Basin using machine learning models. The authors compare and analyze the performance of three machine learning of random forest, linear kernel support vector machine, and radial basis function support vector machine. The results show that the random forest algorithm outperforms the other algorithms in terms of accuracy. There are some concerns in the manuscript. Below are my comments. I hope the authors find them useful.
1. The authors should clarify why the three machine learning models are used for debris flow susceptibility modelling.
2. The generation of debris flow inventory should be clarified.
3. It would be better to provide a flowchart in the methodology part.
4. The susceptibility map generated by linear SVM appears quite unusual, with almost no areas classified as very low and low susceptibility. The author should provide an explanation for this anomaly.
5. Why is the Conclusion section placed before the Discussion section?
6. The authors used grid search for hyperparameter optimization. It would be better to include the search ranges for different models in the manuscript?

Experimental design

see basic reporting

Validity of the findings

no comment

Additional comments

no comment

---

## Round 0.2 · Minor Revisions

· Academic Editor

Minor Revisions

Dear Dr Zhou,
I've read the final revised manuscript version and found that the reviewers' comments have been addressed carefully.

I've done additional comments that will require a minor munuscirpt revision. Please see the attached pdf, for the details.

Reviewer 1 has requested that you cite specific references. You may add them if you believe they are especially relevant. However, I do not expect you to include these citations, and if you do not include them, this will not influence my decision.

Warm regards,
Mauro Rossi
Academic Editor
PeerJ Life & Environment

Reviewer 1 ·

Basic reporting

Dear Editor,‎

I hope this message finds you well. I extend my sincere appreciation for providing me ‎with the opportunity to revise the manuscript titled “Machine Learning Approaches to ‎Debris Flow Susceptibility Analyses in the Yunnan section of the Nujiang River Basin”, ‎which was submitted to the “PeerJ”. While the manuscript contains intriguing ‎information, there are several modifications and clarifications that require attention. I ‎have outlined these issues below for your consideration:‎
‎-‎ The background provides a clear context for the study, focusing on the Yunnan ‎section of the Nujiang River Basin.‎
‎-‎ Consider adding a brief statement on the significance of studying debris flow ‎susceptibility in alpine-valley areas and the potential implications for hazard ‎mitigation.‎
‎-‎ Clearly state the objectives of the study. What specific goals are the authors ‎aiming to achieve in analyzing debris flow susceptibility in the Yunnan section?‎
‎-‎ Define the scope of the research to manage reader expectations.‎
‎-‎ Commend the choice of three machine learning algorithms (RF, Linear SVM, ‎RBFSVM) for modeling debris flow susceptibility.‎
‎-‎ Provide a concise overview of each algorithm's strengths and applications in ‎debris flow studies.‎
‎-‎ Highlight the significance of the 20 factors considered in determining debris ‎flow susceptibility.‎
‎-‎ Clarify the rationale behind selecting these specific factors and how they ‎contribute to the overall analysis.‎
‎-‎ Clearly present the comparative performance of RF, RBFSVM, and Linear ‎SVM in terms of accuracy. Include quantitative metrics to support the claims.‎
‎-‎ Emphasize the key factors that were identified as critical to debris flow ‎occurrence. Provide insights into how these factors interact and contribute to ‎susceptibility.‎
‎-‎ Summarize the main findings succinctly. For instance, if RF outperforms the ‎other algorithms, state this explicitly and discuss the implications.‎
‎-‎ Discuss the role of topographic conditions, geology, precipitation, vegetation, ‎and anthropogenic factors in debris flow susceptibility.‎
‎-‎ Clearly indicate which factors were identified as dominant in debris flow ‎occurrence.‎
‎-‎ Provide insights into why the relative elevation difference is considered the ‎most prominent evaluation factor.‎
‎-‎ Discuss the construction and interpretation of susceptibility maps based on RF. ‎What do these maps reveal about the distribution of susceptibility zones along ‎the Nujiang River mainstream?‎
‎-‎ Elaborate on how the study contributes to methodological guidance for debris ‎flow susceptibility assessment.‎
‎-‎ Discuss the potential applications of the study's findings and how they could ‎inform hazard mitigation strategies.‎
‎-‎ Summarize the key takeaways from the study.‎
‎-‎ Consider providing recommendations for future research or potential ‎applications of the findings.‎
‎-‎ Ensure that the language used is clear and concise. Avoid ambiguous ‎statements and provide necessary definitions for technical terms.‎
‎-‎ Proofread for grammatical accuracy and coherence in sentence structure.‎
‎-‎ I would like to suggest following scientific articles that will helps authors to ‎improve their manuscript quality and increase level of the work:‎

https://doi.org/10.3390/w16030380
https://doi.org/10.3390/land12071397
https://doi.org/10.1007/s12665-022-10603



Thank you once again for the chance to contribute to the “PeerJ.”‎

Best regards,

Experimental design

see above comments

Validity of the findings

see above comments

Additional comments

see above comments

---

## Round 0.3 · Minor Revisions

· Academic Editor

Minor Revisions

Dear Authors,
Thanks for the work you have done on the manuscript that has been greatly improved after the revision.

There are still two following minor issues to solve:

1) In line 37 at the beginning of the introduction you mention that DF is a particular type of flood, but DFs are traditionally considered by the vast amount of international literature as gravitative processes/landslides. Please refer to largely recognized references and modify the introduction accordingly (e.g., you may refer to Varnes, 1978 or Cruden and Varnes, 1996, or Hungr et al, 2014).

2) In the rebuttal letter (answer to comment MR14) you mention that you have modified table 2 and figure 5 to account for resolutions and scale of input data, but I was not able to find those modifications. I underline again that those are fundamental information to indicate and discuss or at least to acknowledge on the manuscript since they can influence largely the results.


Warm regards,
Mauro Rossi
Academic Editor
PeerJ Life & Environment

---

## Round 0.4 · accepted · Accept

· Academic Editor

Accept

Dear Dr. Zhou and co-authors,

I've reviewed the modifications done to the manuscript and I would like to thank you for having addressed all my comments. The manuscript can be accepted as it is and it can be considered ready for publication.

Regards
Mauro Rossi